# Development of an intervention for reducing infant bathing frequency

Lucy P. Goldsmith[1,2], Michael R. Perkin[1], Charlotte Wahlich[1], Lakshmi Chandrasekaran[1], Victoria Cornelius[3], Robert J. Boyle[4], Carsten Flohr[5], Amanda Roberts[6], Kathryn Willis[1], Michael Ussher[1,7] *

1 Population Health Research Institute, St. George's University of London, London, United Kingdom, 2 School of Health and Psychological Sciences, City, University of London, London, United Kingdom, 3 Imperial Clinical Trials Unit, School of Public Health, Imperial College, London, United Kingdom, 4 National Heart and Lung Institute, Imperial College London, London, United Kingdom, 5 Unit for Paediatric and Population-Based Dermatology Research, St John's Institute of Dermatology, Guy's & St Thomas' NHS Foundation Trust and King's College London, London, United Kingdom, 6 Nottingham Support Group for Carers of Children with Eczema, Nottingham, United Kingdom, 7 Institute of Social Marketing and Health, University of Stirling, Stirling, United Kingdom

* mussher@sgul.ac.uk

## Abstract

### Background

Bathing babies less frequently and intensively in the first six months of life may prevent eczema, but this has not yet been definitively tested in a randomised controlled trial. Such a trial would require evidence-based support to help parents engage with a minimal bathing routine. The present study reports the development of this support.

### Methods

We adopted a four-stage design process: (i) Pregnant women and their families (n = 31) were interviewed to ascertain key barriers and facilitators towards following the minimal bathing intervention. (ii) These barriers and facilitators were mapped to behaviour change techniques, focussing on the intervention types of education, persuasion and environmental restructuring, alongside appropriate modes of delivery, and prototype intervention materials were developed. (iii) We iteratively refined these materials in a workshop with multidisciplinary experts and Patient and Public Involvement and Engagement (PPIE) representatives (n = 13) and an (iv) intervention walkthrough with families (n = 5). The design process was informed by the Behaviour Change Wheel, Theoretical framework of acceptability and the Template for intervention description and replication.

### Results

Social influences and motivational factors are likely to influence both uptake and adherence to the intervention. Anticipated emotional reward from participating in research for the benefit of others was indicated to be a strong facilitator for intervention uptake. Alternatives to bathing, having fun with the baby and the night-time routine, alongside family support, were notable facilitators suggested to aid adherence to the intervention. Barriers included hygiene

**Data Availability Statement:** Most of the relevant data are within the Results section of the paper, in the form of excerpts of the transcripts. Additional data is available in an online repository, in the form of an interactive mapping diagram, illustrating

connections between different elements of the data: https://doi.org/10.24376/rd.sgul.23966574

**Funding:** This study is funded by the National Institute for Health and Care Research; Research for Patient Benefit programme (NIHR203170); https://fundingawards.nihr.ac.uk/award/NIHR203170. MP, RB, CF, MU, VC and AR were co-applicants for funding. The views expressed are those of the author(s) and not necessarily those of the NIHR or the Department of Health and Social Care. The funders had no role in study design, data collection and analysis, decision to publish, or preparation of the manuscript.

**Competing interests:** The authors have declared that no competing interests exist.

concerns and anticipated negative social appraisal. Barriers and facilitators were mapped to thirty-six behaviour change techniques, focussing on the intervention types of education, persuasion and environmental restructuring, all of which were embedded into the package of support. The prototype intervention materials received positive feedback from the expert workshop and study walkthrough with families. The final package of support comprises printed and digital prompts and cues, a study booklet, video, and digital tool for self-monitoring.

## Conclusions

The intervention design process incorporated the 'real world' views and experiences of families, experts and PPIE representatives, alongside criteria for designing behavioural interventions. The effectiveness of the package of support will be tested in a feasibility trial and embedded process evaluation.

## Introduction

Eczema has the highest global burden of all skin diseases, affecting sixteen to thirty percent of children in England [1], with increased prevalence in children from ethnically minoritised groups [1]. Twenty-percent of children worldwide are affected by eczema [2, 3] and the onset of eczema is usually within the first year of life [4]. Childhood eczema is typically associated with sleep disturbance and behavioural difficulties, which exacerbate the distress of eczema, leading to significant impacts on both children and parents [5]. Consequently, primary prevention is extremely important. Excessive bathing agitates the skin of people with established eczema [6] and a large study of healthy babies [7] indicated an association between frequent bathing at three months and signs of impaired skin barrier function [8]. Therefore, an effective primary prevention measure may be to bathe babies less often. The World Health Organization guidelines recommend simply delaying bathing babies for the first 24 hours of life [9]. Guidance from healthcare providers in the UK is extremely varied, with less than 30% of NHS Trusts recommending any delay to the first bath and just under 40% of healthcare providers recommending a bathing frequency, ranging from minimal bathing to advising daily bathing [10]. A previous qualitative study explored views about the infant bathing practices of 20 midwives, 10 heath visitors and 26 women with children (22 with antenatal and postnatal data). They concluded that there is 'informed uncertainty', a term used to reflect that interviewees had information from a variety of sources and their personal beliefs, but that this information challenged the national recommendations, creating doubt. Bathing frequency is not mentioned in the paper reporting the qualitative study's findings [11].

Before a definitive randomised controlled trial can establish whether primary prevention through reduced bathing frequency can be achieved, an evidenced-based intervention is needed to support reduced bathing frequency of new infants. The present study reports the development of a minimal bathing intervention (primarily asking parents to bathe their baby once a week or less for the first six months of the baby's life), using a four-stage design process incorporating parent, expert and patient perspectives, design criteria and theories of behaviour change. The study is part of a larger project, 'BabyBathe' (NIHR203170) [12, 13], which will test the feasibility of the intervention in a randomised controlled study.

## Methods

We used a person-based approach to intervention development [14], with four stages: (i) interviews with pregnant women and their families to identify potential barriers and facilitators to engagement with the intervention, (ii) mapping of the barriers and facilitators identified in (i) to behaviour change techniques (BCTs) and potential modes of delivery, to produce draft intervention materials, (iii) a workshop with stakeholders to assess the acceptability of the draft materials, and (iv) a Patient and Public Involvement and Engagement (PPIE) workshop to refine the intervention materials ready for testing in a feasibility trial. Favourable ethical opinion was given by the North of Scotland Research Ethics Committee on 05/09/22 (22/NS/0120). Methods are reported following the consolidated criteria for reporting qualitative research (COREQ) guidance [15] (see S1 Table)

### Stage 1: Interviews/focus groups with families

We recruited pregnant women at St George's Hospital, London, UK, which provides healthcare for one of the most multi-ethnic populations in the UK, with around 5000 deliveries annually. A researcher or research midwife approached women in the antenatal ultrasound clinic waiting room, to confirm study eligibility, explain the study and provide participant information sheets and consent forms to the woman, her partner, and other family members or child carers potentially interested in participating. For women agreeing to be contacted, a researcher called the woman/family, in the next week, to confirm participation and arrange an interview. At the start of the interview, verbal consent from all participants was recorded digitally and on a consent form by the researcher in accordance with our research ethics approval. Verbal consent was selected to best facilitate consenting several family members from home into the study. Consent forms are available as S1 File.

Eligible women were aged at least 16 years, at least 20 weeks into a singleton, uncomplicated pregnancy, understood English and did not have any condition that would make the use of emollient inadvisable or not possible. This was an exclusion criteria as it could indicate a skin health problem. There is evidence that families with only one child are more likely to bathe their infant daily or more [16] therefore we purposively sampled for those with and without other children. We also purposively sampled for families with and without atopy ("Have you, your partner or any of your children ever been diagnosed with eczema by a doctor?"). We additionally sought maximum variation in family characteristics that might affect bathing behaviours, including age, socio-economic status and ethnicity. We used the 'ten plus three' rule for data saturation [17]—aiming for 10 families with children and 10 without (both 5:5 with and without atopy). Semi-structured interviews/focus groups used a topic guide (see S2 File), allowing participants to speak freely. The number of participants attending interview/focus group varied; for simplicity, we will refer to all of these as interviews. Interviews were conducted remotely by one researcher (LG) via telephone, digitally recorded and transcribed. All participating families were offered a £40 e-gift voucher as a reimbursement for their time. The topic guide was informed by the Theoretical Domains Framework (TDF) [18] and the 'Template for Intervention Description and Replication' (TIDieR) framework for reporting interventions [19]. The Capability-Opportunity-Motivation-Behaviours (COM-B) model was also used to prompt questions about barriers and facilitators related to capability, opportunity and motivation towards minimal bathing. The study materials were developed with two pregnant women and a PPIE representative on the study team.

Thematic analysis [20] of the data was both deductive (informed by topic guide and theories), and inductive, from participants' accounts, and was led by LG. Five researchers (LG, KW, LC, CW, MU) reviewed a selection of transcripts and developed a coding framework.

Using NVivo12, two researchers independently coded 20% of randomly selected transcripts (LG, MU), the coding framework was revised, then all the transcripts were coded. Themes were developed through discussion with the wider team. To indicate the frequency of themes we use the terms "all", "almost all", "most", "the majority", "some", and "a few".

## Stage 2: Use of behavioural change techniques and development of intervention materials

A list of potential barriers and facilitators to intervention engagement were extracted from stage 1 interview data. We then reviewed the standard taxonomy of 93 BCTs [21] and selected all BCTs which might potentially target the identified barriers and facilitators, focussing on the intervention types of education, persuasion and environmental restructuring. We then mapped each of these BCTS to potential modes of delivery. Based on these mappings, we drafted intervention materials and materials for the control group, ready to present in the stage 3 workshop.

## Stage 3: Stakeholder workshop

We held a workshop, in-person, at St George's, University of London, purposively sampling for a range of stakeholders. At the workshop we summarised the findings from stages 1 and 2, then reviewed the proposed modes of delivery for each BCT. We presented draft intervention materials, including a website, information booklet, and study prompts. To aid decision-making, we presented items from the APEASE criteria (affordability, practicability, effectiveness and cost-effectiveness, acceptability, side effects/safety and equity) [22] and Theoretical Framework of Acceptability (TFA) criteria (affective attitude, burden, perceived effectiveness, ethicality, intervention coherence, opportunity costs, and self-efficacy) [23]. We also asked participants to consider whether any aspect of the intervention should be tailored to different contexts or groups. Participants were asked to email any additional comments. We circulated revised materials to participants for additional comments. Notes were made concerning participants' comments, which were checked by participants.

## Stage 4: PPIE walkthrough/focus group

We conducted a walkthrough/focus group with families (target sample n = 8) to pilot and refine the intervention materials. Six women were newly recruited and two women who participated in the first stage qualitative interviews and have a baby were also included. Eligibility criteria and recruitment were the same as stage 1. The walkthrough was led by three researchers and conducted online. Questions and prompts were informed by the Theoretical Framework of Acceptability [23]. Using the prototype intervention materials, participants underwent walkthroughs in real-life scenarios (e.g., bedtime routines) to illustrate how the BCTs would work. The walkthrough was audio recorded and transcribed. Data from the walkthroughs were analysed inductively from participant's accounts and deductively using the TFA [14]. Results were used to finalise the intervention.

### Reflexive note

Ultimately, we are investigating whether there is a causal association between frequency of bathing infants and the development of eczema, and agreement that there is currently no evidence to supports such a link. Two members of the study team (including the PPI expert) have lived experience of eczema. Most members of the study team are White British (White Other (n = 1), Asian (n = 1)), and all members have lived in the UK for most of their lives. Stage 1

interviews and data analysis were led by LG, a female postdoctoral researcher with substantial experience in qualitative research. To minimise any bias that may be introduced by our beliefs, authors reflected about any potential impacts on the research and endeavoured to remain objective. All participants were recruited by author LG. Participants knew that author LG is a researcher working on the study and keen to listen to their perspective to ensure the intervention is 'really user-friendly'. LG has personal experience of eczema and no children.

## Results

### Stage 1: Interviews/focus groups

**Stage 1 recruitment.** Forty-five pregnant women discussed participation, completed screening forms and provided contact details. Twenty women consented to participate alongside consenting family members. The study was open to recruitment for stage 1 between October 26th 2022 and February 27th 2023. Of these families, thirteen had history of atopy; seven of which had children and six were expecting their first child. Of seven families without atopy history, four already had children and three were expecting their first child. Thirty-one participants were recruited, including twenty pregnant women, ten fathers and one sister of an expectant mother. Five participants identified as Asian or Asian British, two as Black, Black British, Caribbean or African, one as mixed ethnicity, fourteen as White British, eight as White Other, and one as 'Other Ethnic Group'.

Pregnant women were between 22 and 39 weeks pregnant (mean (SD) = 35.2 (4.5) weeks). Ages of these women ranged from 28–42 years (mean (SD) = 33.9 (3.6)). Fifteen participants were educated to degree level.

**Stage 1 data analysis: Barriers and facilitators to the intervention.** *1.1. Social influences*: *Family and social circle.* Almost all parents considered wider social influences as important. Some felt that, in their social networks, the intervention would not be considered unusual:

". . .advice our parents and also from our friends who'd had babies. And then they just said, they're not really that dirty so you don't want to irritate the skin, if they're not really, really dirty you don't have to wash them every day. So we just followed the advice of the people around us, from their experience." (Mother, BBWP1-41)

A few parents reported that despite their wider family norms differing substantially from the study intervention, they expected their family to respect and support their decision to follow the intervention:

"If we said . . . this is what we're doing because this is what we think is right, I think they would respect what we say. I don't think it would be hugely a thing, you know, for them telling us what we're doing with our own child and how often we're bathing them and stuff like that. They just want to be involved. . ." (Mother, BBWP1-36)

A few parents reported differing perspectives within the family. Where parents anticipated criticism due to following the intervention, some reported they would avoid this by not divulging how often they bathed their baby.

"Mum: I think my mum would think it was probably. . .would think it was dirty. Your parents would probably be okay about it.

Dad: They'd probably just say it's interesting or quirky.

Mum: I think it would be okay, I don't think that anyone needs to know how often you're bathing a baby." (Parents, BBWP1-12)

Talking about the aim of the intervention, to reduce infant eczema, was anticipated to make it easier to gain support:

"I will explain how they told that I need to try it, because they need to find some solution for...my baby for eczema." (Mother, BBWP1-30)

A few participants anticipated that sharing the study aims would also help them gain support for study participation, even if they were in the control condition:

"...neither me nor my sister had eczema but my cousin had it really badly. So I think she (the maternal grandmother) would be pleased that it might help other people." (Mother, BBWP1-31)

Most parents did not expect criticism from their social circle:

"I've got some friends that have had children and my sister... So I think I'd ask them for advice... But in terms of influence, I think it would just be, you know, more like helpful advice rather than opinions..." (Mother, BBWP1-36)

When asked how often they were planning to bathe the baby in the first few weeks and months, a few parents considered infrequent bathing to be a social norm:

"Once a week or twice a week at the most... from the antenatal class... Because we don't want to irritate the baby's skin." (BBWP1-26)

In contrast, some parents reported that nightly bathing is a norm:

"But now all of the Bump and Baby crew they all bath their babies every night. So, it is part of their routine." (Mother, BBWP1-05)

Almost all ethnic minority interviewees with darker skin reported that, within their culture, bathing a baby only once a week or less would be considered strange:

"It might be a little bit trickier for them (the grandparents) because, especially, they're from an Asian background, keeping the baby clean and moisturised is very important to the health of the baby, so it might take a bit of education to get them onboard" (Mother, BBWP1-41)

"Obviously...different cultures might do things differently, but for me, that is part of my life, that is having a bath every day... it's just standard, like I come from a Jamaican culture" (Mother, BBWP1-07)

*1.2 Social influences*: *Study team and health professionals.* We sought parent's advice about the optimum time to approach potential participants about the study. All parents considered late pregnancy to be most suitable because at that point parents make plans for the baby's arrival.

"I've just had my... midwife appointment today (36 weeks) and it could easily have been something a midwife mentioned in that... I think by mentioning it too early in pregnancy your brain is full of pregnancy, not actually after the birth." (Mother, BBWP1-16)

Almost all parents felt that a professional with practical childcare skills, rather than a researcher, would be most suitable for providing information about the intervention.

"I think, yes, I think maybe someone who already deals with babies. So, a health visitor or a midwife." (Mother, BBWP1-02)

Some parents suggested that it would be helpful to be able to contact the study team with queries, such as what to do if they're struggling to follow the intervention or to get their families 'on board':

"I need to convince one or two of my relatives for this. Maybe my husband's sisters. . . because they think bathing is really, really important to the baby, keep cleaning. So I don't know how to convince them. . .I need some advice" (Mother, BBWP1-30)

Most parents also reported that written information, including about situations where they might not want to follow the intervention (e.g.; if baby develops a skin problem) would be helpful. Written information was also suggested to help explain the intervention to others.

"I think it needs to be something that there's something provided just so that they can understand 'cause you know what it's like, you've told me and then I'll give them a version of what you've told me. . .." (Mother, BBWP1-28)

*1.3 Social Influences*: *Concerns about social judgements from family and professionals*. One participant said her husband was alarmed by the idea of bathing their baby only once a week or less:

"You remember, he's like, what? We're bathing the baby, what are you talking about?" (BBWP1-07)

And that her mother would have a very negative attitude to the idea:

"My mum would have been absolutely shocked to hear you say once a week, yeah, she would have been, no way. . . My mum would have asked me if I was crazy" (BBWP1-07)

A few other parents reported that they expected difficulties with grandparents:

"I'd have no problem telling them but my mum would. . . find it ridiculous, wouldn't she? She wouldn't be able to understand it." (Mother, BBWP1-05)

*2.1 Motivational factors*. For all interviewees, anticipated consequences of bathing were very motivational for participating in a trial. These included health consequences, such as doing something to reduce the chances of the infant developing eczema, and reward from knowing that they are contributing to science for the benefit of many.

"Like I said, the main question in my mind now is like, does it actually bring on eczema in someone. I'd like to know earlier if you can just prevent it happening entirely, kind of thing. So, and if that is the case, then there's huge benefits." (Mother, BBWP1-02)

Parents who have an older child with eczema were particularly likely to comment on potential health benefits:

"It, kind of, made me want to try and see, with this new baby, if I don't bath it as often, to see if she'll end up getting eczema or not, her skin will be better or worse than (the older sibling)." (Mother, BBWP1-41)

Improved environmental consequences via reduced usage of energy, water and plastics were anticipated by some:

". . .an environmental impact is probably less bathwater that's getting, you know. . . Less chemical products that are being used, less plastic bottles. . ." (Mother, BBWP1-31)

Convenience and time saving was anticipated by a few parents, alongside topping and tailing being easier than bathing:

"Less work. . . I guess as first time parents it was quite daunting to bath, you know, a newborn baby, . . .I guess that we had to prepare a lot of stuff, we'd have to make sure the room was warm, the temperature's nice and then you have to make sure that we dry them properly. And it is extra stuff that you have to do in your day, so the advantage of bathing less is easier on the mums." (Mother, BBWP1-41)

However, some other parents thought a bath could be easier at times and that they wouldn't see saving time as a benefit.

"If they've had a big poo, I suppose. It might be easier to stick them in a bath rather than try and do a, kind of, top and tail." (Mother, BBWP1-35)

"I don't find the bath inconvenient. I'd put that as a really lovely time" (Mother, BBWP1-05)

"Yeah, I guess what you said about them having more time, I guess they will have more time because they're not running a bath, putting them in, playing with them, getting them dried afterwards. Like that takes up time. I think from our point of view that's a guaranteed half an hour. . . actually a guaranteed hour where we're not on our phones. . . So, you know, you'll have your time back but what are you going to do with that time?" (Father, BBWP1-05)

Many interviewees also anticipated negative emotional consequences, such as missing the shared experience of bathing which could support bonding, and the baby missing out on the enjoyment.

"it might be something that really soothes the baby, and they might like being in the water, and it might be a calming thing. . . it might be something that becomes part of bedtime routine, just a signal, it's night time now. . .If I know that the baby really likes bath time. . . it's going to be very tempting to want to do that more." (Mother, BBWP1-16)

Some had hygiene concerns about cleaning a baby thoroughly without a bath:

"Ah, disadvantages. I guess, like, more irritation to the skin because you aren't. . .even maybe with the cotton ball you might not be able to clean it thoroughly. Especially on newborns it's hard to, I find that it's very scary to move their joins or arms or necks, you

know. . . But in the bath you're able to splash water and that's easier and more thorough." (Mother, BBWP1-41)

In contrast, some parents did not have strong hygiene concerns:

"But I do think that once a week when they're really new, it just seems to, sort of, make logical sense. . . as long as you're, you know, obviously keeping them clean, like, with nappy changes and little top and tail, like, strip wash or whatever you call it, I don't really see why it would be necessary, they're not going to smell bad." (Mother, BBWP1-36)

*3.1 Goals and planning*: *Behaviour substitution.* The need for behaviour substitution was a key theme. Three types of substitution were mentioned. Firstly, a few reported using topping and tailing as an alternative to bathing in the past:

"Yes, it was (an alternative to bathing), I still liked to clean his bottom and his face because I noticed that if I don't top and tail him and then dry him really well he does get nappy rash. And then because his face is just dirty, clean him, yeah" (Mother, BBWP1-41)

Secondly, almost all parents suggested the need for substitutions for bonding with and having fun with the baby:

". . .he did enjoy bathing because it was really relaxing and afterwards we do a bit of baby massage, so whenever we did have a bath it really calmed him down." (Mother, BBWP1-41)

Alternatives for the father to bond with the baby were perceived as particularly important:

"I don't know, dads, that you can turn nappy changing into a bonding period. So, I think one of the advantages with the bath time is that you get to relax with them, you can put some toys in and it's a playtime or it's a bonding time and that'll be important. . . So, if you reduce that, then you reduce those opportunities. But that's not to say you couldn't find them obviously in other areas." (Father, BBWP1-22)

Thirdly, some parents felt that bathing was an integral part of the night-time routine:

"I think it's just from my, sort of, well, my learned knowledge of knowing that, like my niece and friend's children, that bath and bedtime routine is really important, doing bath, bed, book, chill. So wanting that to become part of a relaxed time to wind-down and make sure they're ready for bed." (Mother, BBWP1-36)

Whereas some parents felt uncertain whether a bath is a necessary part of this routine:

"I guess what I don't know is whether a bath, is it just for cleaning or is it. . . does it become part of the routine, does it help calm them down or is it something that they hate? I think I might also just see based on the baby if they even enjoy a bath and then how much you end up doing that. And if it's an ordeal every time, then maybe I'd do less." (Mother, BBWP1-31)

A few parents who had not included a bath as part of the night-time routine with older siblings suggested alternatives:

"I think we just skipped it (the bath) and I think we just added in an extra story. If we had a bit of time that we wanted to fill before bed, so bath is only ten minutes, it's not like an adult bathing. And we did that instead almost nightly." (Mother, BBWP1-13)

A few first-time parents had plans for alternatives to bathing:

"I think maybe, like, instead of it being a bath, like, have that final nappy change with that, sort of, little cleaning routine, and then they're into their PJs and then it's, you know, turn the lights down low, have a little night light, read a book, real chill, do a little book, even though they're, you know, probably not really interested in (a bath) at that age." (Mother, BBWP1-36)

*3.2 Goals and planning*: *Monitoring skincare activities*. We asked for feedback about self-monitoring to help parents follow the intervention. This was positively received by all parents—especially where the responsibility for skincare is anticipated to be shared between parents, as it can help parents work together.

"I think an app would be ideal to actually record this information, and it's also good for yourself, because you'll be able to see it first hand, so that you can see it in real time. . . I could do it every day, because once a week, you're going to forget. . . (Mother, BBWP1-07)

*3.3 Goals and planning*: *Willingness and what to do when finding it difficult to follow the intervention regime.* Nearly all parents indicated that they needed guidance about what to do if they wanted to bathe their baby more frequently, such as on a very hot day, exceedingly messy nappies, the baby developing skin problems or if the parents are finding it difficult to establish an effective night-time routine without bathing. The need for flexibility in the intervention was emphasised.

". . .as long as it's on the understanding that if they need to have a bath, they've got to have a bath, you know, if they've had a massive poonami explosion." (Mother, BBWP1-28)

**Potential intervention components.**    *4.1 Social support*: *Facilitating partner involvement.* Ensuring that both partners could participate in taking responsibility for following the intervention was important to a few couples. It was important that any tools designed to support self-monitoring (such as an app) were user-friendly and suitable for multiple adults.

"Like I said, I do need to discuss it with (the father) first, but I think if he understands that it's because there's potential that it might reduce the chance of getting eczema and it's for a wider study that's generally helpful to other babies, I think he'd be happy to use it (use the app)." (Mother, BBWP1-28)

Some women felt that they'd like to discuss the study with their partner and any other child carers before participating, but a few women were clear that the final decision about joining the study would be theirs'.

". . . it's only myself and my partner that are, you know, going to be in charge of that in terms of, like, even if someone's looking after her, we'll be the ones that say she needs a bath tonight or she doesn't need a bath tonight." (Mother, BBWP1-36)

*4.2 Social support from midwives*. Meeting with a midwife to discuss the study before the birth was considered to be useful by most parents, with instruction about how to follow the intervention, provide an opportunity to ask questions and to rehearse the bathing routine, bolster self-belief and gain emotional support. This could include a discussion and watching a video together. Midwives' practical knowledge and skills was highly valued.

"I would say somebody with experience of babies. . . so, they can answer any practical questions. So, from my point of view I'd say a midwife." (BBWP1-28)

*4.3 Social support from the study team*. Social support from the study team on an ad-hoc basis was perceived to be valuable by some participants to help them handle any uncertainty about following the intervention.

". . .you know when their bums are so sore, wiping them is really unkind. It hurts them when you wipe. And when that used to happen, we used to spray her down with the bath shower hose. I don't know if that counts as bathing. . ." (Mother, BBWP1-13)

Many parents assumed they would have to 'drop out' of the study if they couldn't follow the intervention, and support from the study team could help retain them:

"I think the only thing obviously I'd worry about is if there was a medical reason why I needed to do it more or less. . . If my child actually had very severe eczema and we were at the GP, but at that point we'd probably drop out the trial. (BBWP1-13)

Where appropriate, on-demand emotional support could also help parents focus on past success, the contribution they are making to science even if they can't follow the intervention, and bolster their self-belief about getting 'back on track' with the intervention.

"You know, just encouraging, it's like someone running a marathon, you just need the cheerleaders at the side there. . . Even just encouragement, we're humans, we're fickle, you get bored easily and, you know." (Father, BBWP1-12)

We sought feedback about a digital reminder from the study team to complete the log (e.g., a text message). This was considered helpful by all parents:

"And the positive thing about say, an app, is that you could do push reminders to people. You haven't submitted a bathing update for two weeks, or whatever, and then they just kick it and go yeah, last Thursday and yesterday. . . I think allowing people to set the frequency of reminders is fine, or frequency of how often they fill it in is fine. . . I reckon your first time mum(s). . . are going to want that daily reminder." (BBWP1-13)

"Yeah, I think probably like a text would be the easiest one. Like the one they'll pay most attention to would be a text or a WhatsApp" (Mother, BBWP1-05)

*4.4 Intervention 'kit': Study booklet/e-booklet*. A few parents suggested a written study booklet could support them to explain the study to others and gain social support for their involvement, or at least help address social barriers to their participation. Some parents thought a booklet would be useful as they were worried about forgetting information.

"I think it needs to be something that there's something provided just so that they can understand 'cause you know what it's like, you've told me and then I'll give them a version of what you've told me" (Mother, BBWP1-28)

Some parents felt it would help motivate their behaviour to know more about the science and range of normal behaviour for infant skincare.

"I think if I knew what was normal. . .what was, like, recommended. . . or, like, if I read a bit more about the benefits of doing it once a week then I think I could be up for it." (Mother, BBWP1-12)

"If it gives scientific information then they might be open to, you know, listen to it, the science, yeah. . .. especially my mum and (the father's) mum both had children who had very bad eczema, so I think they would be intrigued to know whether it does make a difference. (Mother, BBWP1-41)

The booklet could support parents with action planning for behaviour substitution as the infant grows over the first six months and different barriers to following the intervention are encountered.

"If we get another terrible sleeper and we were going to fall into that trap of would this help her sleep?" (Mother, BBWP1-13)

Mum: "I think that I'd go in confident and then it'd get quite hard after like 12 weeks." Dad: "Yeah. That's what I think as well. I think the same thing. I think it's easy to agree in principle but we're first time parents, so you just never know what is going to happen and what you're going to rely on, what you're going to need. What you think is the right thing to do, it could be very different now to what you experience when he's actually here. . ." (BBWP1-16)

*4.5 Intervention 'kit': Prompts, cues and study-related equipment.* We sought feedback about a study aide-memoire in the bathroom. Feedback from all parents was positive, particularly from mums with older children who thought it could help them avoid falling back on habitual behaviours. They also said the prompt/cue should be waterproof.

"Maybe, like a little. . . something that goes on the tiles or something that's maybe like a visual aid, as a reminder for, like, bath temperature should be this much, your bathing days are these days. . . Even like a. . . like a magnet or just something that's laminated, like, waterproof on a sucker which you could stick on the bath." (Mother, BBWP1-36)

"I think it would be useful, especially if it had something that you could also write on. You could collect your data as you go along." (Mother, BBWP1-35)

Most parents also felt that electronic cues would be useful:

"Yeah, I think, yeah, I think so, some sort of visual aid that's either on your phone as a reminder or, like, physically as a reminder. . .." (Mother, BBWP1-36)

A few parents felt that if the study asked them to check the temperature of the bath, then a thermometer should be provided:

"I think you probably have to give people a bath thermometer. . . Or offer them one. Do you want one? And if they've got one already fine. If they don't, great. Because it's a thing that has a price." (Mother, BBWP1-13)

*4.6 Intervention 'kit': Self-monitoring diary*. Keeping a diary of skincare activities was thought to facilitate co-ordination between caregivers. It was felt important that the diary is easy and quick to use and most participants preferred an app:

". . .an app would be great. . . I just feel like an app is so quick. . . Honestly, parents monitor everything. . ." (BBWP1-22)

An online diary was considered to be more time-consuming by all participants. Almost all parents who commented about a paper diary disliked the idea as they thought it may be misplaced and wouldn't be practical:

"I'm just thinking App because, I like pen and paper, I'm quite old school like that, it's just they have a habit of getting lost or just by definition if you want to record the information when you're in the bathroom they're likely to get soggy and ripped and they're just not very practical." (BBWP1-28)

Having lots of options about when to record skincare activities was valued by participants–some felt they would be likely to log the data immediately, others at the end of the day or week:

"I think just leaving it so they can put the information in whenever they like" (Mother, BBWP1-42)

"I would just, I'd say, do it when you're doing it, because especially if you're bathing your baby more than once a week, you're just going to forget what you've done and what you've used. . . I think, definitely a weekly reminder is helpful, but also just having a portal or some sort of, like, functionality to do it whenever you want. . ." (Mother, BBWP1-36)

Onerous data collection could put off some parents:

"I think I feel confident now but I'm not trying to look after a small human, but I guess it would depend on how often you needed information and all that sort of stuff." (Mother, BBWP1-36)

## Stage 2: Use of behavioural change techniques and development of intervention materials

The stage two results, of mapping the barriers and facilitators to BCTs then mapping the BCTs to modes of delivery, were iteratively refined throughout the design process and the final mapping is presented in an interactive diagram [24].

## Stage 3: Stakeholder workshop

The stakeholder workshop, held on May 5th 2023, included multidisciplinary experts (n = 13) from dermatology (1), paediatrics (1), behavioural science and intervention design (1), health

visiting (3), midwifery (2), allergic disease (1), general practice (1), health psychology (1) and psychology (1), alongside a PPIE representative with experience of eczema.

The intervention materials were well received–they were described by one workshop attendee in feedback after the meeting as "overwhelmingly positive and suggested changes very minor." Attendees suggested that we enhance 'credibility' of the information in the booklet by adding official logos and convey information about the expertise of the study team and the way the study was designed. Using the logo for the app in the booklet (and on prompts/cues) was suggested to help participants remember the app icon for data entry. Even though swimming is not part of the intervention, an attendee suggested that the app should monitor swimming.

The suggestion to use 'waterwipes' (baby cleaning wipes with no preservatives, just water) for cleaning under the nappy area, as in the draft booklet, was criticised by a midwife as there are concerns that these wipes are not hygienic as the packet can become contaminated as soon as it is opened. There was a discussion regarding the phrase 'keeping baths short' in the booklet. Participants considered whether parents would ask what this means in terms of minutes of bathing. However, it was agreed that it would be better to leave this open for individual interpretation. We discussed the booklet's advice about cleaning the nappy area, of "making sure to clean in all folds in the skin", or whether this could be considered to be patronising. Midwives and health visitors all advised that this level of detail was needed. One attendee suggested that we ask for study participants to provide contact details for a close friend or family member who we can contact if we cannot get in touch with them. The discussion also covered monitoring the risk of cross-contamination through the monthly questionnaire. Participants also suggested translating the study materials into languages spoken by the local population (e.g., Tamil or Hindi) and to mention breastfeeding in the section entitled 'Would bathing less affect bonding with my baby?'. The discussion also included how to best structure the planned study video to incorporate relevant BCTs. It is currently unclear whether frequent bathing of babies increases the risk of eczema, therefore it was agreed that this association should be presented with caution. The potential ease of delivering the intervention in the NHS was discussed. Consequently, it was considered that 'social support from midwives' should be minimised and covered in the booklet and/or video. Similarly, it was agreed that much of the 'social support from the study team' could be covered in the booklet. It was also decided that prompts to complete the study e-monitoring diary could be digital, rather than requiring direct contact with the study team.

## Stage 4: PPIE walkthrough/focus group

Prior to the walkthrough/focus group, we asked participants to download and experiment with the study app and visit the study website, including watching the online video and reading the intervention booklet. The study walkthrough was held online on 6[th] July 2023. Attendees included two newly recruited women, alongside two women from stage one. An adapted one-to-one workshop was held with a further participant who was unable to attend on the original date. All participants already had children; three had a family history of atopy and two did not. One was Asian or Asian British, one Black or Black British, one White British, one White Other, and one 'Other Ethnic Group'. They were between 22 and 34 weeks pregnant and aged ranged 24 to 32 years and four were educated to degree level. Recruitment was open between July 3[rd] to July 12[th] 2023.

The study materials were well received by participants and perceived to be attractively presented, clear, and relatable. One participant reported that requiring the passcode to be entered every time the app opens was annoying and this was changed. Participants suggested providing

more information about eczema in the study booklet and app, including the prevalence and information about severe eczema. We decided against this as we didn't want to over-inflate the risks of eczema for participants and a causal association between bathing and eczema is uncertain. One participant suggested placing the booklet on the website, with easy to navigate section titles which expand to reveal the information, as opposed to having a downloadable pfd. and we have taken this forward along with the suggestion that important information is highlighted. Another suggestion we adopted is to incorporate the MyCap study app logo into a welcome email as some participants initially found the standard MyCap app logo confusing when they tried to download the app before the meeting. The focus group echoed previous feedback about monthly milestone messages sent through the app, along with other reminders.

## Final intervention

The final intervention materials include, for both groups, a study booklet which clarifies how to participate in the intervention and, for the intervention group, incorporates BCTs (see S3 and S4 Files). Both groups will receive a study bookmark and fridge magnet as prompts/cues for entering study information. The intervention group will additionally be provided with an online study video which supports the delivery of twenty BCTs, a bath thermometer, wallchart and waterproof bathroom visual aid/prompt which deliver additional BCTs including self-monitoring and restructuring the physical environment as outlined in the interactive mapping diagram [24]. The interactive mapping diagram comprehensively illustrates how the barriers and facilitators are matched to BCTs incorporated in the intervention and the BCTs are embedded in each mode of delivery. It is available to download freely from an online repository [24]. Images of the pack contents are available in S1 and S2 Figs. In the intervention researchers are providing personal contact with participants but if the study were rolled out this might be provided by health professionals.

## Discussion

This study reports a four-stage intervention development process which employs theory-driven approaches to develop materials for an intervention which primarily aims to reduce bathing of infants to once a week or less for the first six month of life. First, in interviews with families, we collected information about barriers and facilitators to uptake and engagement with the intervention and gathered suggestions about acceptable modes of intervention delivery. Then we mapped the identified barriers and facilitators to potential BCTs and mode of delivery and developed prototype materials. In the penultimate stage we reviewed and refined the prototype materials with clinical and PPIE experts. In the final stage we consulted with families, in a structured walkthrough, to verify the acceptability of the intervention materials and finalise these materials ready for the feasibility trial. Throughout this process we considered whether design aspects would be equally suitable for different cultural groups and whether any tailoring would be beneficial. Comprehensive insight into external challenges and constraints of "real world" contexts was integrated into the design in stages one and four from families and in stage three from clinical experts. Throughout the process we relied on input from a PPIE expert study team member.

Parents indicated many barriers to following the intervention, including hygiene concerns and anticipated negative social appraisal as a consequence of following the intervention. Alongside the barriers, many facilitators were also identified. Anticipated emotional reward from participating in research to help others is perhaps the primary facilitator for deciding to take part in the study. Alternatives for bathing, having fun with the baby and the night-time

routine alongside family support were facilitators suggested to help participants adhere to the intervention. Thirty-six BCTs were built into the intervention from the comprehensive scoping. Perhaps the most important of these are credible source, goal setting, action planning and prompts and cues, each of which was prominent in the qualitive work and is delivered in multiple ways. The stakeholder workshop clarified our thinking that the BCTs selected should be non-coercive, focussing on education, persuasion and environmental restructuring, and helping participants feel good about being in the study. This philosophy of BCT delivery is particularly suitable for research where there is equipoise regarding whether either study condition is beneficial (i.e., it is unknown whether the frequency of bathing babies impacts on risk of eczema developing). The family participants strongly supported the idea of an app to facilitate self-monitoring. We have designed an intervention which requires very low input from staff in order that it is suitable for large-scale roll-out in the NHS, if shown to be effective and cost effective by a definitive trial.

A strength of this study is the extended, four-stage design process, incorporating the views of patients and clinical and patient experts, combined with the in-depth use of both behaviour change and intervention design theories. A potential limitation is that the pregnant women who were interviewed at stage one were well-educated, with 75% having a degree. Within London the proportion of adults with a degree is 50% [25]; higher than other parts of the country. In Wandsworth, the borough in which the recruiting hospital is located, this rises to 63% [25]. Considering the generational trend in the UK, in which younger people are more likely to have degrees, the recruited population may well be reflective of the local population of pregnant women. It is also possible that more educated women were more interested in participating. This highly educated population, living in the urban, culturally diverse environment of this London borough, may have different attitudes towards the intervention compared with pregnant women in other areas and with less education.

## Conclusions

In conclusion, the intervention has been developed through an extensive four-stage process incorporating the views of families and stakeholders, including their insights into "real world" contexts and the suitability of the intervention for different cultural groups. The final intervention implements thirty-six behaviour change techniques, focussing on education, persuasion and environmental restructuring, across eight modes of delivery. Ultimately, the success of the design will be indicated by the level of behaviour change achieved in the BabyBathe feasibility trial [12], which is now underway.

## Supporting information

**S1 Table. Consolidated criteria for reporting qualitative research (COREQ) items.**
(DOCX)

**S1 File. Participant informed consent forms.**
(PDF)

**S2 File. Interview topic guides.**
(DOCX)

**S3 File. Study booklet for intervention group.**
(PDF)

**S4 File. Study booklet for control group.**
(PDF)

**S1 Fig. Image of intervention group pack.**
(JPEG)

**S2 Fig. Image of control group pack.**
(JPG)

## Acknowledgments

We would like to thank the Centre of Evidence Based Dermatology (CEBD) Patient Panel at the University of Nottingham for their input into this work.

## Author Contributions

**Conceptualization:** Michael R. Perkin, Robert J. Boyle, Carsten Flohr, Amanda Roberts, Michael Ussher.

**Data curation:** Lucy P. Goldsmith.

**Formal analysis:** Lucy P. Goldsmith, Charlotte Wahlich, Lakshmi Chandrasekaran, Kathryn Willis, Michael Ussher.

**Funding acquisition:** Michael R. Perkin, Victoria Cornelius, Robert J. Boyle, Carsten Flohr, Amanda Roberts, Michael Ussher.

**Investigation:** Lucy P. Goldsmith.

**Methodology:** Lucy P. Goldsmith, Michael Ussher.

**Project administration:** Lucy P. Goldsmith, Michael R. Perkin.

**Software:** Lucy P. Goldsmith.

**Supervision:** Michael R. Perkin, Michael Ussher.

**Validation:** Lucy P. Goldsmith, Charlotte Wahlich, Michael Ussher.

**Visualization:** Michael Ussher.

**Writing – original draft:** Lucy P. Goldsmith, Michael Ussher.

**Writing – review & editing:** Lucy P. Goldsmith, Michael R. Perkin, Charlotte Wahlich, Lakshmi Chandrasekaran, Victoria Cornelius, Robert J. Boyle, Carsten Flohr, Amanda Roberts, Kathryn Willis, Michael Ussher.

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
