## [Decision Letter · Decision Letter 0]

28 Nov 2023

PONE-D-23-26320Development of an intervention for reducing infant bathing frequencyPLOS ONE

Dear Dr. Goldsmith,

Thank you for submitting your manuscript to PLOS ONE. After careful consideration, we feel that it has merit but does not fully meet PLOS ONE’s publication criteria as it currently stands. Therefore, we invite you to submit a revised version of the manuscript that addresses the points raised during the review process.

We look forward to receiving your revised manuscript.

Kind regards,

Ivan Sarmiento

Academic Editor

PLOS ONE

Journal Requirements:

2. Please note that in order to use the direct billing option the corresponding author must be affiliated with the chosen institute. Please either amend your manuscript to change the affiliation or corresponding author, or email us at plosone@plos.org with a request to remove this option.

Reviewers' comments:

Reviewer's Responses to Questions

**Comments to the Author**

1. Is the manuscript technically sound, and do the data support the conclusions?

Reviewer #1: Yes

2. Has the statistical analysis been performed appropriately and rigorously? 

Reviewer #1: Yes

3. Have the authors made all data underlying the findings in their manuscript fully available?

Reviewer #1: Yes

4. Is the manuscript presented in an intelligible fashion and written in standard English?

Reviewer #1: Yes

5. Review Comments to the Author

Reviewer #1: The article's proposal, entitled "Development of an intervention for reducing infant bathing frequency", is very well presented and corresponds to public health research with scientific relevance, with a concrete and realistic question that can have practical and effective application in health prevention.

The project was duly approved by an ethical opinion issued by the North of Scotland Research Ethics Committee. I want to ask why this was done with a committee in Scotland and not with a committee in London or St George's Hospital, London.

The authors adequately present the Consolidated Criteria for Reporting Qualitative Research (COREQ) items.

The methodology is well structured, clearly presenting the inclusion criteria and intervention steps to recognize the likely positive and negative factors influencing acceptance to participate in an evaluation study. Interestingly, the authors state, "Our epistemological stance in this study was critical realism – which assumes that experiences are understood through human interpretation and mediated by our beliefs and perceptions”.

The results show a precise sequence of stages and the inclusion of feedback in developing the intervention materials. Indeed, as the authors rightly state, the intervention design process incorporated the "real world" views and experiences of families, experts, and PPIE representatives.

The abstract's conclusions do not state a significant result for behaviour change techniques: education, persuasion, and restructuring of the environment. These techniques should be included in the abstract's conclusions.

6. PLOS authors have the option to publish the peer review history of their article (what does this mean?). If published, this will include your full peer review and any attached files.

Reviewer #1: **Yes: **Germán Zuluaga Ramírez

---

## [Author Response · Author response to Decision Letter 0]

19 Jan 2024

We have attached the document 'Response to reviewers', in which we respond to the reviewer and editor comments

---

## [Editor Report · Decision Letter 1]

23 Jan 2024

Development of an intervention for reducing infant bathing frequency

PONE-D-23-26320R1

Dear Dr. Ussher,

We’re pleased to inform you that your manuscript has been judged scientifically suitable for publication and will be formally accepted for publication once it meets all outstanding technical requirements.

Kind regards,

Ivan Sarmiento

Academic Editor

PLOS ONE
---

## [Editor Report · Acceptance letter]

16 Feb 2024

PONE-D-23-26320R1 

PLOS ONE

Dear Dr. Ussher, 

I'm pleased to inform you that your manuscript has been deemed suitable for publication in PLOS ONE. Congratulations! Your manuscript is now being handed over to our production team.

Kind regards, 

on behalf of

Dr. Ivan Sarmiento 

Academic Editor

PLOS ONE